# PREVENTING POSTERIOR COLLAPSE WITH $\delta$-VAES

**Ali Razavi**
Deepmind
alirazavi@google.com

**Aäron van den Oord**
Deepmind
avdnoord@google.com

**Ben Poole**
Google Brain
pooleb@google.com

**Oriol Vinyals**
Deepmind
vinyals@google.com

## ABSTRACT

Due to the phenomenon of "posterior collapse," current latent variable generative models pose a challenging design choice that either weakens the capacity of the decoder or requires altering the training objective. We develop an alternative that utilizes the most powerful generative models as decoders, optimize the variational lower bound, and ensures that the latent variables preserve and encode useful information. Our proposed $\delta$-VAEs achieve this by constraining the variational family for the posterior to have a minimum distance to the prior. For sequential latent variable models, our approach resembles the classic representation learning approach of slow feature analysis. We demonstrate our method's efficacy at modeling text on LM1B and modeling images: learning representations, improving sample quality, and achieving state of the art log-likelihood on CIFAR-10 and ImageNet $32 \times 32$.

## 1 INTRODUCTION

Deep latent variable models trained with amortized variational inference have led to advances in representation learning on high-dimensional datasets (Kingma & Welling, 2013; Rezende et al., 2014). These latent variable models typically have simple decoders, where the mapping from the latent variable to the input space is unimodal, for example using a conditional Gaussian decoder. This typically results in representations that are good at capturing the global structure in the input, but fail at capturing more complex local structure (e.g. texture (Larsen et al., 2016)). In parallel, advances in autoregressive models have led to drastic improvements in density modeling and sample quality without explicit latent variables (van den Oord et al., 2016b). While these models are good at capturing local statistics, they often fail to produce globally coherent structures (Ostrovski et al., 2018).

Combining the power of tractable densities from autoregressive models with the representation learning capabilities of latent variable models could result in higher-quality generative models with useful latent representations. While much prior work has attempted to join these two models, a common problem remains. If the autoregressive decoder is expressive enough to model the data density, then the model can learn to ignore the latent variables, resulting in a trivial posterior that collapses to the prior. This phenomenon has been frequently observed in prior work and has been referred to as *optimization challenges* of VAEs by Bowman et al. (2015), the *information preference* property by Chen et al. (2016), and the *posterior collapse* problems by several others (e.g. van den Oord et al., 2017; Kim et al., 2018; Dieng et al., 2018). Ideally, an approach that mitigates posterior collapse would not alter the evidence lower bound (ELBO) training objective, and would allow the practitioner to leverage the most recent advances in powerful autoregressive decoders to improve performance. To the best of our knowledge, no prior work has succeeded at this goal. Most common approaches either change the objective (Higgins et al., 2017; Alemi et al., 2017; Zhao et al., 2017; Chen et al., 2016; Lucas & Verbeek, 2017; Goyal et al., 2017), or weaken the decoder (Bowman et al., 2015; Gulrajani et al., 2016). Additionally, these approaches are often challenging to tune and highly sensitive to hyperparameters (Alemi et al., 2017; Chen et al., 2016).

In this paper, we propose $\delta$-VAEs, a simple framework for selecting variational families that prevent posterior collapse without altering the ELBO training objective or weakening the decoder. By restricting the parameters or family of the posterior, we ensure that there is a minimum KL divergence, $\delta$, between the posterior and the prior.

We demonstrate the effectiveness of this approach at learning latent-variable models with powerful decoders on images (CIFAR-10, and ImageNet $32 \times 32$), and text (LM1B). We achieve state of the art log-likelihood results with image models by additionally introducing a sequential latent-variable model with an anti-causal encoder structure. Our experiments demonstrate the utility of $\delta$-VAEs at learning useful representations for downstream tasks without sacrificing performance on density modeling.

## 2 MITIGATING POSTERIOR COLLAPSE WITH $\delta$-VAES

Our proposed $\delta$-VAE builds upon the framework of variational autoencoders (VAEs) (Kingma & Welling, 2013; Rezende et al., 2014) for training latent-variable models with amortized variational inference. Our goal is to train a generative model $p(x, z)$ to maximize the marginal likelihood $\log p(x)$ on a dataset. As the marginal likelihood requires computing an intractable integral over the unobserved latent variable $z$, VAEs introduce an encoder network $q(z|x)$ and optimize a tractable lower bound (the ELBO): $\log p(x) \geq \mathbb{E}_{z \sim q(z|x)} [\log p(x|z)] - D_{\mathrm{KL}}(q(z|x)\|p(z))$. The first term is typically referred to as the reconstruction term, and the second term (KL) the rate term, as it measures how many nats on average are required to send the latent variables from the encoder $(q(z|x))$ to the decoder $(p(z|x))$ using a code designed for the prior $(p(z))$ (Hoffman et al., 2016; Alemi et al., 2017).

The problem of posterior collapse is that the rate term, $D_{\mathrm{KL}}(q(z|x)\|p(z))$ reduces to 0. In this case, the approximate posterior $q(z|x)$ equals the prior $p(z)$, thus the latent variables do not carry any information about the input $x$. A necessary condition if we want representations to be meaningful is to have the rate term be positive.

In this paper we address the posterior collapse problem with structural constraints so that the KL divergence between the posterior and prior is lower bounded by design. This can be achieved by choosing families of distributions for the prior and approximate posterior, $p_\theta(\mathbf{z})$ and $q_\phi(\mathbf{z}|\mathbf{x})$ such that $\min_{\theta,\phi} D_{\mathrm{KL}}(q_\phi(\mathbf{z}|\mathbf{x})\|p_\theta(\mathbf{z})) \geq \delta$. We refer to $\delta$ as the *committed rate* of the model.

Note that a trivial choice for $p$ and $q$ to have a non-zero committed rate is to set them to Gaussian distributions with fixed (but different) variance term. We study a variant of this case in the experiments, and provide more details of this setup in Appendix D. In the following section we describe our choices for $p_\theta$ and $q_\phi$, but others should also be explored in future work.

### 2.1 $\delta$-VAE WITH SEQUENTIAL LATENT VARIABLES

Data such as speech, natural images and text exhibit strong spatio-temporal continuity. Our aim is to model variations in such data through latent variables, so that we have control over not just the global characteristics of the generated samples (e.g., existence of an object), but also can influence their finer, often shifting attributes such as texture and pose in the case of natural images, tone, volume and accent in the case of speech, or style and sentiment in the case of natural language. Sequences of latent variables can be an effective modeling tool for expressing the occurrence and evolution of such features throughout the sequence.

To construct a $\delta$-VAE in the sequential setting, we combine a mean field posterior with a correlated prior in time. We model the posterior distribution of each timestep as $q(\mathbf{z}_t|\mathbf{x}) = \mathcal{N}(\mathbf{z}_t; \mu_t(\mathbf{x}), \sigma_t(\mathbf{x}))$. For the prior, we use a first-order linear autoregressive process (AR(1)), where $\mathbf{z}_t = \alpha \mathbf{z}_{t-1} + \epsilon_t$ with $\epsilon_t$ zero mean Gaussian noise with constant variance $\sigma_\epsilon^2$. The conditional probability for the latent variable can be expressed as $p(\mathbf{z}_t|\mathbf{z}_{<t}) = \mathcal{N}(\mathbf{z}_t; \alpha \mathbf{z}_{t-1}, \sigma_\epsilon)$. This process is wide-sense stationary (that is, having constant sufficient statistics through its time evolution) if $|\alpha| < 1$. If so, then $\mathbf{z}_t$ has zero mean and variance of $\frac{\sigma_\epsilon^2}{1-\alpha^2}$. It is thus convenient to choose $\sigma_\epsilon = \sqrt{1 - \alpha^2}$ so that the variance is constant over time. The mismatch in the correlation structure of the prior and the posterior results in the following positive lower bound on the KL-divergence between the two distributions (see Appendix C for derivation):

$$D_{\mathrm{KL}}(q(\mathbf{z}|\mathbf{x})\|p(\mathbf{z})) \geq \frac{1}{2}\sum_{k=1}^{d}(n-2)\ln(1+\alpha_k^2) - \ln(1-\alpha_k^2) \tag{1}$$

where $n$ is the length of the sequence and $d$ is the dimension of the latent variable at each timestep. The committed rate between the prior and the posterior is easily controlled by equating the right hand side of the inequality in equation 1 to a given rate $\delta$ and solving for $\alpha$. In Fig. 1, we show the scaling of the minimum rate as a function of $\alpha$ and the behavior of $\delta$-VAE in 2d.

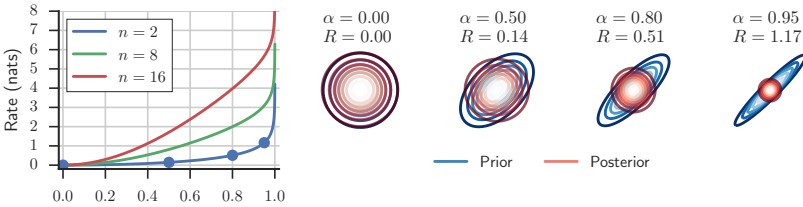

Figure 1: *Effect of $\delta$ in a toy model.* Fitting an uncorrelated Gaussian for the posterior, $q_\phi(z)$, to a correlated Gaussian prior, $p_\alpha(z)$, by minimizing $D_{\mathrm{KL}}(q_\phi(z)\|p_\alpha(z))$ over $\phi$. Left: committed rate ($\delta$) as a function of the prior squared correlation $\alpha$ and the dimensionality $n$. Right: contours of the optimal posterior and prior in 2d. As the correlation increases, the minimum rate grows.

### 2.1.1 Relation to Probabilistic Slowness Prior

The AR(1) prior over the latent variables specifies the degree of temporal correlation in the latent space. As the correlation $\alpha$ approaches one, the prior trajectories get smoother . On the other hand in the limit of $\alpha$ approaching 0, the prior becomes the same as the independent standard Gaussian prior where there are no correlations between timesteps. This pairing of independent posterior with a correlated prior is related to the probabilistic counterpart to Slow Feature Analysis (Wiskott & Sejnowski, 2002) in Turner & Sahani (2007). SFA has been shown to be an effective method for learning invariant spatio-temporal features (Wiskott & Sejnowski, 2002). In our models, we infer latent variables with multiple dimensions per timestep, each with a different slowness filter imposed by a different value of $\alpha$, corresponding to features with different speed of variation.

### 2.2 Anti-Causal Encoder Network

Having a high capacity autoregressive network as the decoder implies that it can accurately estimate $p(\mathbf{x}_t|\mathbf{x}_{<t})$. Given this premise, what kind of complementary information can latent variables provide? Encoding information about the past seems wasteful as the autoregressive decoder has full access to past observations already. On the other hand, if we impose conditional independence between observations and latent variables at other timesteps given the current one (i.e., $p(\mathbf{x}_t|\mathbf{z}) = p(\mathbf{x}_t|\mathbf{z}_t)$), there will then be at best (by the data processing inequality (Cover & Thomas, 2006)) a break-even situation between the KL cost of encoding information in $\mathbf{z}_t$ and the resulting improvement in the reconstruction loss. There is therefore no advantage for the model to utilize the latent variable even if it would transmit to the decoder the unobserved $\mathbf{x}_t$. The situation is different when $\mathbf{z}_t$ can inform the decoder at multiple timesteps, encoding information about $\mathbf{x}_t$ and $\mathbf{x}_{>t}$. In this setting, the decoder pays the KL cost for the mutual information once, but is able to leverage the transmitted information multiple times to reduce its entropy about future predictions.

To encourage the generative model to leverage the latents for future timesteps, we introduce an *anti-causal* structure for the encoder where the parameters of the variational posterior for a timestep cannot depend on past observations (Fig. 2). Alternatively, one can consider a *non-causal* structure that allows latents be inferred from all observations. In this non-causal setup there is no temporal order in either the encoder or the decoder, thus the model resembles a standard non-temporal latent-variable model. While the anti-causal structure is a subgraph of the non-causal structure, we find that the anti-causal structure often performs better, and we compare both approaches in different settings in Appendix F.1.

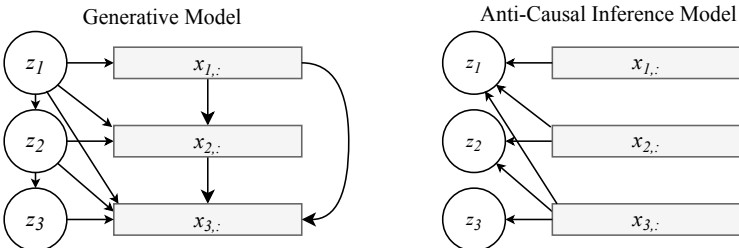

Figure 2: Generative structures for the inference of sequential latent variables. The anti-causal structure introduces an inductive bias to encode in each latent variable information about the future

## 3 RELATED WORK

The main focus of our work is on representation learning and density modeling in latent variable models with powerful decoders. Earlier work has focused on this kind of architecture, but has addressed the problem of posterior collapse in different ways.

In terms of our architecture, the decoders for our image models build on advances in autoregressive modeling from van den Oord et al. (2016b); Salimans et al. (2017); Chen et al. (2017); Parmar et al. (2018). Unlike prior models, we use sequential latent variables to generate the image row by row. This differs from Gregor et al. (2016), where the latent variables are sequential but the entire image is generated at each timestep. Our sequential image generation model resembles latent variable models used for timeseries (Chung et al., 2015; Babaeizadeh et al., 2017; Denton & Fergus, 2018) but does not rely on KL annealing, and has an additional autoregressive dependence of the outputs over time (rows of the image). Another difference between our work and previous sequential latent variable models is our proposed anti-causal structure for the inference network (see Sect. 2.2). We motivate this structure from a coding efficiency and representation learning standpoint and demonstrate its effectiveness empirically in Sect. 4. For textual data, we use the Transformer architecture from Vaswani et al. (2017) as our main blueprint for the decoder. As shown in Sect. 4, our method is able to learn informative latent variables while preserving the performance of these models in terms of likelihoods.

To prevent posterior collapse, most prior work has focused on modifying the training objective. Bowman et al. (2015); Yang et al. (2017); Kim et al. (2018) and Gulrajani et al. (2016) use an annealing strategy, where they anneal the weight on the rate from 0 to 1 over the course of training. This approach does not directly optimize a lower bound on likelihood for most of training, and tuning the annealing schedule to prevent collapse can be challenging (see Sect. 4). Similarly, Higgins et al. (2017) proposes using a fixed coefficient $> 1$ on the rate term to learn disentangled representations. Zhao et al. (2017) adds a term to the objective to pick the model with maximal rate. Other works use auxiliary tasks such as secondary low-resolution reconstructions with non-autoregressive decoders (Lucas & Verbeek, 2017) or predicting the state of the backward LSTM in the encoder (Goyal et al., 2017) to encourage the utilization of latent variables. Chen et al. (2016); Kingma et al. (2016) use free-bits to allow the model to hit a target minimum rate, but the objective is non-smooth which leads to optimization difficulties in our hands, and deviations from a lower bound on likelihood when the soft version is used with a coefficient less than 1. Alemi et al. (2017) argue that the ELBO is a defective objective function for representation learning as it does not distinguish between models with different rates, and advocate for model selection based on downstream tasks. Their method for sweeping models was to use $\beta$-VAE with different coefficients, which can be challenging as the mapping from $\beta$ to rate is highly nonlinear, and model- and data-dependent. While we adopt the same rate-distortion perspective as Alemi et al. (2017), we present a new way of achieving a target rate while optimizing the vanilla ELBO objective.

Most similar to our approach is work on constraining the variational family to regularize the model. VQ-VAE (van den Oord et al., 2017) uses discrete latent variables obtained by vector quantization of the latent space that, given a uniform prior over the outcome, yields a fixed KL divergence equal to $\log K$, where $K$ is the size of the codebook. A number of recent papers have also used the von Mises-Fisher (vMF) distribution to obtain a fixed KL divergence and mitigate the posterior

collapse problem. In particular, Guu et al. (2017); Xu & Durrett (2018); Davidson et al. (2018) use $vMF(\mu, \kappa)$ with a fixed $\kappa$ as their posterior, and the uniform distribution (*i.e.* $vMF(\cdot, 0)$) as the prior. The mismatching prior-posterior thus give a constant KL divergence. As such, this approach can be considered as the continuous analogue of VQ-VAE. Unlike the VQ-VAE and vMF approaches which have a constant KL divergence for every data point, $\delta$-VAE can allow higher KL for different data points. This allows the model to allocate more bits for more complicated inputs, which has been shown to be useful for detecting outliers in datasets (Alemi et al., 2018). As such, $\delta$-VAE may be considered a generalisation of these fixed-KL approaches.

The Associative Compression Networks (ACN) (Graves & Menick, 2014) is a new method for learning latent variables with powerful decoders that exploits the associations between training examples in the dataset by amortizing the description length of the code among many similar training examples. ACN deviates from the i.i.d training regime of the classical methods in statistics and machine learning, and is considered a procedure for compressing whole datasets rather than individual training examples. GECO (Jimenez Rezende & Viola, 2018) is a recently proposed method to stabilize the training of $\beta$-VAEs by finding an automatic annealing schedule for the KL that satisfies a tolerance constraint for maximum allowed distortion, and solving the resulting Lagrange multiplier for the KL penalty. The value of $\beta$, however, does not necessarily approach one, which means that the optimized objective may not be a lower bound for the marginal likelihood.

## 4 EXPERIMENTS

### 4.1 NATURAL IMAGES

We applied our method to generative modeling of images on the CIFAR-10 (Krizhevsky et al.) and downsampled ImageNet (Deng et al., 2009) ($32 \times 32$ as prepared in van den Oord et al. (2016a)) datasets. We describe the main components in the following. The details of our hyperparameters can be found in Appendix E.

**Decoder**: Our decoder network is closest to PixelSNAIL (Chen et al., 2017) but also incorporates elements from the original GatedPixelCNN (van den Oord et al., 2016b). In particular, as introduced by Salimans et al. (2017) and used in Chen et al. (2017), we use a single channel network to output the components of discretised mixture of logistics distributions for each channel, and linear dependencies between the RGB colour channels. As in PixelSNAIL, we use attention layers interleaved with masked gated convolution layers. We use the same architecture of gated convolution introduced in van den Oord et al. (2016b). We also use the multi-head attention module of Vaswani et al. (2017). To condition the decoder, similar to Transformer and unlike PixelCNN variants that use 1x1 convolution, we use attention over the output of the encoder. The decoder-encoder attention is causally masked to realize the anti-causal inference structure, and is unmasked for the non-causal structure.

**Encoder**. Our encoder also uses the same blueprint as the decoder. To introduce the anti-causal structure the input is reversed, shifted and cropped by one in order to obtain the desired future context. Using one latent variable for each pixel is too inefficient in terms of computation so we encode each row of the image with a multidimensional latent variable.

**Auxiliary Prior**. Tomczak & Welling (2017); Hoffman et al. (2016); Jimenez Rezende & Viola (2018) show that VAE performance can suffer when there is a significant mismatch between the prior and the aggregate posterior, $q(z) = \mathbb{E}_{x \sim \mathcal{D}}[q(z|x)]$. When such a gap exists, the decoder is likely to have never seen samples from regions of the prior distribution where the aggregate posterior assigns small probability density. This phenomenon, also known as the "posterior holes" problem (Jimenez Rezende & Viola, 2018), can be exacerbated in $\delta$-VAEs, where the systematic mismatch between the prior and the posterior might induce a large gap between the prior and aggregate posterior. Increasing the complexity of the variational family can reduce this gap (Rezende & Mohamed, 2015), but require changes in the objective to control the rate and prevent posterior collapse (Kingma et al., 2016). To address this limitation, we adopt the approaches of van den Oord et al. (2017); Roy et al. (2018) and train an auxiliary prior over the course of learning to match the aggregate posterior, but that does not influence the training of the encoder or decoder. We used a simple autoregressive model for the auxiliary prior $p^{aux}$: a single-layer LSTM network with conditional-Gaussian outputs.

### 4.1.1 DENSITY ESTIMATION RESULTS

We begin by comparing our approach to prior work on CIFAR-10 and downsampled ImageNet 32x32 in Table 1. As expected, we found that the capacity of the employed autoregressive decoder had a large impact on the overall performance. Nevertheless, our models with latent variables have a negligible gap compared to their powerful autoregressive latent-free counterparts, while also learning informative latent variables. In comparison, (Chen et al., 2016) had a 0.03 bits per dimension gap between their latent variable model and PixelCNN++ architecture[1]. On ImageNet 32x32, our latent variable model achieves on par performance with purely autoregressive Image Transformer (Parmar et al., 2018). On CIFAR-10 we achieve a new state of the art of 2.83 bits per dimension, again matching the performance of our autoregressive baseline. Note that the values for KL appear quite small as they are reported in bits per dimension (e.g. 0.02 bits/dim translates to 61 bits/image encoded in the latents). The results on CIFAR-10 also demonstrate the effect of the auxiliary prior on improving the efficiency of the latent code; it leads to more than $50\%$ (on average 30 bits per image) reduction in the rate of the model to achieve the same performance.

| | CIFAR-10 Test | ImageNet $32 \times 32$ Valid |
|---|---|---|
| **Latent Variable Models** | | |
| ConvDraw (Gregor et al. (2016)) | $\leq 3.85$ | - |
| DenseNet VLAE (Chen et al. (2016)) | $\approx 2.95$ | - |
| $\delta$-VAE + PixelSNAIL + AR(1) Prior | $\leq 2.85$ (0.02) | $\leq 3.78$ (0.08) |
| $\delta$-VAE + PixelSNAIL + Auxiliary Prior | $\leq$ **2.83** (0.01) | $\leq$ **3.77** (0.07) |
| **Autoregressive Models** | | |
| Gated PixelCNN(van den Oord et al. (2016b)) | 3.03 | 3.83 |
| PixelCNN++ (Salimans et al. (2017)) | 2.92 | - |
| PixelRNN (van den Oord et al. (2016a)) | 3.00 | - |
| ImageTransformer (Parmar et al. (2018)) | 2.90 | **3.77** |
| PixelSNAIL (Chen et al. (2017)) | 2.85 | 3.80 |
| Our Decoder baseline | **2.83** | **3.77** |

Table 1: Estimated upper bound on negative log-likelihood along with KL-divergence (in parenthesis) in bits per dimension for CIFAR-10 and downsampled ImageNet.

### 4.2 UTILIZATION OF LATENT VARIABLES

In this section, we aim to demonstrate that our models learn meaningful representations of the data in the latent variables. We first investigate the effect of $\mathbf{z}$ on the generated samples from the model. Fig. 3 depicts samples from an ImageNet model (see Appendix for CIFAR-10), where we sample from the decoder network multiple times conditioned on a fixed sample from the auxiliary prior. We see similar global structure (e.g. same color background, scale and structure of objects) but very different details. This indicates that the model is using the latent variable to capture global structure, while the autoregressive decoder is filling in local statistics and patterns.

For a more quantitative assessment of how useful the learned representations are for downstream tasks, we performed linear classification from the representation to the class labels on CIFAR-10. We also study the effect of the chosen rate of the model on classification accuracy as illustrated in Fig. 4b, along with the performance of other methods. We find that generally a model with higher rate gives better classification accuracy, with our highest rate model, encoding 92 bits per image, giving the best accuracy of $68\%$. However, we find that improved log-likelihood does not necessarily lead to better linear classification results. We caution that an important requirement for this task is the linear separability of the learned feature space, which may not align with the desire to learn highly compressed representations.

### 4.3 ABLATION STUDIES

We performed more extensive comparisons of $\delta$-VAE with other approaches to prevent posterior collapse on the CIFAR-10 dataset. We employ the same medium sized encoder and decoder for

---

[1]the exact results for the autoregressive baseline was not reported in Chen et al. (2016)

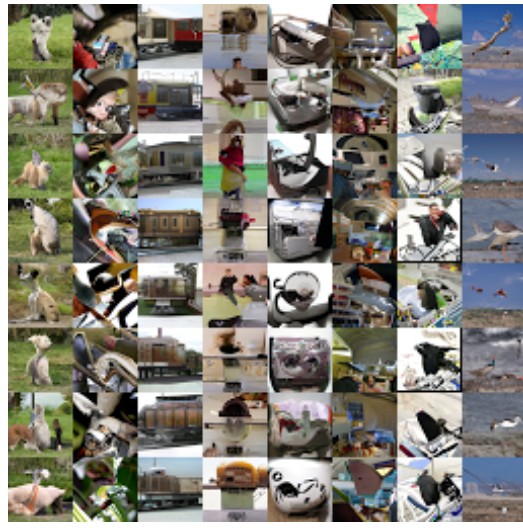 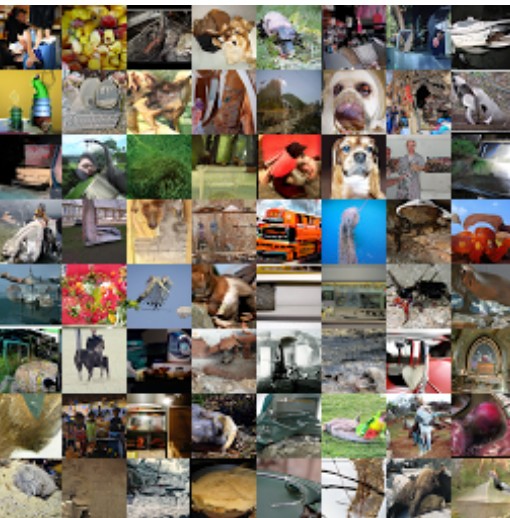

(a) Multiple decoding of the same **z**     (b) Random samples from the our Auxiliary prior

Figure 3: Random samples from our ImageNet $32 \times 32$ model. Each column in Fig. 3a shows multiple samples from $p(\mathbf{x}|\mathbf{z})$ for a fixed $\mathbf{z} \sim p_{aux}(\mathbf{z})$. Each image in Fig. 3b is decoded using a different sample from $p_{aux}(\mathbf{z})$.

evaluating all methods as detailed in Appendix E. Fig. 4a reports the rate-distortion results of our experiments for the CIFAR-10 test set. To better highlight the difference between models and to put into perspective the amount of information that latent variables capture about images, the rate and distortion results in Fig. 4a are reported in bits per images. We only report results for models that encode a non-negligible amount information in latent variables. Unlike the committed information rate approach of $\delta$-VAE, most alternative solutions required considerable amount of effort to get the training converge or prevent the KL from collapsing altogether. For example, with linear annealing of KL (Bowman et al., 2015), despite trying a wide range of values for the end step of the annealing schedule, we were not able to train a model with a significant usage of latent variables; the KL collapsed as soon as $\beta$ approached one. A practical advantage of our approach is its simple formula to choose the target minimum rate of the model. Targeting a desired rate in $\beta$-VAE, on the other hand, proved to be difficult, as many of our attempts resulted in either collapsed KL, or very large KL values that led to inefficient inference. As reported in Chen et al. (2016), we also observed that optimising models with the free-bits loss was challenging and sensitive to hyperparameter values.

To assess each methods tendency to overfit across the range of rates, we also report the rate-distortion results for CIFAR-10 training sets in Appendix F. While $\beta$-VAEs do find points along the rate-distortion optimal frontier on the training set, we found that they overfit more than $\delta$-VAEs, with $\delta$-VAEs dominating the rate-distortion frontier on heldout data.

Next, we compare the performance of the anti-causal encoder structure with the non-causal structure on the CIFAR-10 dataset discussed in Sect. 2.2. The results for several configurations of our model are reported in the Appendix Table 6. In models where the decoder is not powerful enough (such as our 6-layer PixelCNN that has no attention and consequently a receptive field smaller than the causal context for most pixels), the anti-causal structure does not perform as well as the non-causal structure. The performance gap is however closed as the decoder becomes more powerful and its receptive field grows by adding self-attention and more layers. We observed that the anti-causal structure outperforms the non-causal encoder for very high capacity decoders, as well as for medium size models with a high rate. We also repeated these experiments with both anti-causal and non-causal structures but without imposing a committed information rate or using other mitigation strategies, and found that neither structure by itself is able to mitigate the posterior collapse issue; in both cases the KL divergence drops to negligible values ($< 10^{-8}$ bits per dimension) only after a few thousand training steps, and never recovers.

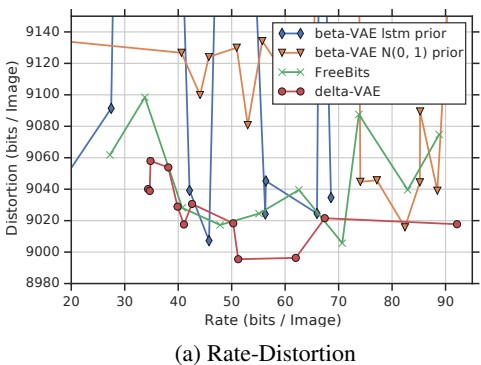

(a) Rate-Distortion

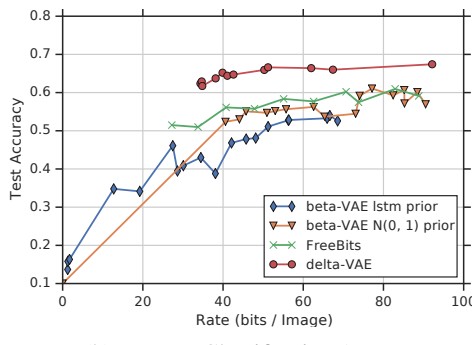

(b) Rate vs. Classification Accuracy

Figure 4: Comparison of CIFAR-10 test performance of $\delta$-VAEs vs. models trained with free-bits and $\beta$-VAE across many rates. $\delta$-VAE is significantly more stable, achieves competitive density estimation results across different rates, and its learned representations perform better in the downstream linear classification task.

## 4.4 TEXT

For our experiments on natural language, we used the 1 Billion Words or LM1B (Chelba et al., 2013) dataset in its processed form in the Tensor2Tensor (Vaswani et al., 2018) codebase [2]. Our employed architecture for text closely follows the Transformer network of Vaswani et al. (2017). Our sequence of latent variables has the same number of elements as in the number of tokens in the input, each having two dimensions with $\alpha = 0.2$ and $0.4$. Our decoder uses causal self-attention as in Vaswani et al. (2017). For the anti-causal structure in the encoder, we use the inverted causality masks as in the decoder to only allow looking at the current timestep and the future.

Quantitatively, our model achieves slightly worse log-likelihood compared to its autoregressive counterpart (Table 2), but makes considerable use of latent variables, as demonstrated by the samples and interpolations in Appendix H.

|  | AR(1) ELBO (KL) | Aux prior ELBO (KL) | AR baseline NLL |
|---|---|---|---|
| $\delta$-VAE | $\leq 3.61(0.2)$ | $\leq 3.58(0.17)$ | 3.55 |

Table 2: The result of our text experiments on LM1B in nats / token.

## 5 DISCUSSION

In this work, we have demonstrated that $\delta$-VAEs provide a simple, intuitive, and effective solution to posterior collapse in latent variable models, enabling them to be paired with powerful decoders. Unlike prior work, we do not require changes to the objective or weakening of the decoder, and we can learn useful representations as well as achieving state-of-the-art likelihoods. While our work presents two simple posterior-prior pairs, there are a number of other possibilities that could be explored in future work. Our work also points to at least two interesting challenges for latent-variable models: (1) can they exceed the performance of a strong autoregressive baseline, and (2) can they learn representations that improve downstream applications such as classification?

ACKNOWLEDGMENTS

We would like to thank Danilo J. Rezende, Sander Dieleman, Jeffrey De Fauw, Jacob Menick, Nal Kalchberner, Andy Brock, Karen Simonyan and Jeff Donahue for their help, insightful discussions and valuable feedback.

---

[2]https://github.com/tensorflow/tensor2tensor

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

## A  DERIVATION OF THE KL DIVERGENCE FOR SEQUENTIAL LATENT VARIABLES

$$
D_{\mathrm{KL}}(q(\mathbf{z}|\mathbf{x})\|p(\mathbf{z})) = \int_{\mathbf{z}} q(\mathbf{z}|\mathbf{x}) \log \frac{q(\mathbf{z}|\mathbf{x})}{p(\mathbf{z})} d\mathbf{z}
$$

$$
= \int_{\mathbf{z}} \prod_{i=1}^{n} q(\mathbf{z}_i|\mathbf{x}) (\sum_{i=1}^{n} \log q(\mathbf{z}_i|\mathbf{x}) - \log p(\mathbf{z})) d\mathbf{z}
$$

$$
= \int_{\mathbf{z}} \prod_{i=1}^{n} q(\mathbf{z}_i|\mathbf{x}) (\sum_{i=1}^{n} \log q(\mathbf{z}_i|\mathbf{x}) - \log p(\mathbf{z}_1) - \sum_{i=2}^{n} \log p(\mathbf{z}_i|\mathbf{z}_{i-1})) d\mathbf{z}
$$

$$
= \int_{\mathbf{z}} \prod_{i=1}^{n} q(\mathbf{z}_i|\mathbf{x}) (\sum_{i=1}^{n} \log q(\mathbf{z}_i|\mathbf{x})) d\mathbf{z} - \int_{\mathbf{z}_1} q(\mathbf{z}_1|\mathbf{x}) \log p(\mathbf{z}_1) d\mathbf{z}_1
$$

$$
- \sum_{i=2}^{i=n} [\int_{\mathbf{z}_{i-1}} q(\mathbf{z}_{i-1}|\mathbf{x}) \int_{\mathbf{z}_i} q(\mathbf{z}_i|\mathbf{x}) \log p(\mathbf{z}_i|\mathbf{z}_{i-1}) d\mathbf{z}_i d\mathbf{z}_{i-1}]
$$

$$
= \int_{\mathbf{z}_1} q(\mathbf{z}_1|\mathbf{x}) \log q(\mathbf{z}_1|\mathbf{x}) d\mathbf{z}_1 - \int_{\mathbf{z}_1} q(\mathbf{z}_1) \log p(\mathbf{z}_1) d\mathbf{z}_1
$$

$$
+ \sum_{i=2}^{n} [\int_{\mathbf{z}_i} q(\mathbf{z}_i|\mathbf{x}) \log q(\mathbf{z}_i|\mathbf{x}) d\mathbf{z}_i - \int_{\mathbf{z}_{i-1}} q(\mathbf{z}_{i-1}|\mathbf{x}) \int_{\mathbf{z}_i} q(\mathbf{z}_i|\mathbf{x}) \log p(\mathbf{z}_i|\mathbf{z}_{i-1}) d\mathbf{z}_i d\mathbf{z}_{i-1}]
$$

$$
= D_{\mathrm{KL}}(q(\mathbf{z}_1|\mathbf{x})\|p(\mathbf{z}_1)) + \sum_{i=2}^{n} [\int_{\mathbf{z}_{i-1}} q(\mathbf{z}_{i-1}|\mathbf{x}) \int_{\mathbf{z}_i} q(\mathbf{z}_i|\mathbf{x}) \log \frac{q(\mathbf{z}_i|\mathbf{x})}{p(\mathbf{z}_i|\mathbf{z}_{i-1})} d\mathbf{z}_i d\mathbf{z}_{i-1}]
$$

$$
= D_{\mathrm{KL}}(q(\mathbf{z}_1|\mathbf{x})\|p(\mathbf{z}_1)) + \sum_{i=2}^{n} \mathbb{E}_{\mathbf{z}_{i-1} \sim q(\mathbf{z}_{i-1}|\mathbf{x})} [D_{\mathrm{KL}}(q(\mathbf{z}_i|\mathbf{x})\|p(\mathbf{z}_i|\mathbf{z}_{i-1}))]
$$

## B  DERIVATION OF THE KL-DIVERGENCE BETWEEN AR(1) AND DIAGONAL GAUSSIAN, AND ITS LOWER-BOUND

$$
\mathbf{z}_i \in \mathbb{R}^d, \mathbf{z}_0 \in \{0\}^d
$$
$$
p(\mathbf{z}_1) = \mathcal{N}(0, 1)
$$
$$
p(\mathbf{z}_i|\mathbf{z}_{i-1}) = \mathcal{N}(\alpha \mathbf{z}_{i-1}, \sqrt{1-\alpha^2}) \quad i > 1
$$
$$
q(\mathbf{z}_i|\mathbf{x}) = \mathcal{N}(\mu_i^\theta(\mathbf{x}), \sigma_i^\theta(\mathbf{x}))
$$

Noting the analytic form for the KL-divergence for two uni-variate Gaussian distributions:

$$
D_{\mathrm{KL}}(\mathcal{N}(\mu_q, \sigma_q)\|\mathcal{N}(\mu_p, \sigma_p)) = \frac{1}{2}[\ln((\frac{\sigma_p}{\sigma_q})^2) + \frac{\sigma_q^2 + (\mu_p - \mu_q)^2}{\sigma_p^2} - 1] \tag{2}
$$

we now derive the lower-bound for KL-divergence. To avoid clutter we assume a single dimension per timestep but extend the results to the general multivariate case at the end of this section.

$$
D_{\mathrm{KL}}(q(\mathbf{z}|\mathbf{x})\|p(\mathbf{z})) = \mathbb{E}_{\mathbf{z}_{i-1} \sim q(\mathbf{z}_{i-1}|\mathbf{x})} [\sum_{i=1}^{n} D_{\mathrm{KL}}(\mathcal{N}(\mu_{q_i}, \sigma_{q_i})\|\mathcal{N}(\mu_{p_i}, \sigma_{p_i}))]
$$

$$
= \frac{1}{2} \mathbb{E}_{\mathbf{z}_{i-1}} [-\ln(\sigma_1^2) + \sigma_1^2 - 1 + \mu_1^2 + \sum_{i=2}^{n} \ln(\frac{1-\alpha^2}{\sigma_i^2}) + \frac{\sigma_i^2}{1-\alpha^2} - 1 + \frac{(\mu_i - \alpha \mathbf{z}_{i-1})^2}{1-\alpha^2}]
$$

$$
= \frac{1}{2} (f_1(\sigma_1^2) + \mu_1^2 + \sum_{i=2}^{n} f_1(\frac{\sigma_i^2}{1-\alpha^2}) + \frac{1}{1-\alpha^2} \mathbb{E}_{\mathbf{z}_{i-1}} [(\mu_i - \alpha \mathbf{z}_{i-1})^2])
$$

Where $f_a(x) = ax - ln(x) - 1$. Using the fact that $\sigma_i^2 = \mathbb{E}[(z_i - \mu_i)^2] = \mathbb{E}(z_i^2) - \mu_i^2$, the expectation inside the summation can be simplified as follows.

$$\mathbb{E}_{z_{i-1}}[(\mu_i - \alpha z_{i-1})^2)]) =$$

$$= \mathbb{E}_{z_{i-1}}[\mu_i^2 - 2\mu_i \alpha z_{i-1} + \alpha^2 z_{i-1}^2]$$
$$= \mu_i^2 - 2\alpha \mu_i \mathbb{E}_{z_{i-1}}[z_{i-1}] + \alpha^2 \mathbb{E}_{z_{i-1}}[z_{i-1}^2]$$
$$= \mu_i^2 - 2\alpha \mu_i \mu_{i-1} + \alpha^2 \sigma_{i-1}^2 + \alpha^2 \mu_{i-1}^2$$
$$= (\mu_i - \alpha \mu_{i-1})^2 + \alpha^2 \sigma_{i-1}^2$$

Plugging this back gives us the following analytic form for the KL-divergence for the sequential latent variable $\mathbf{z}$.

$$D_{\mathrm{KL}}(q(\mathbf{z}|\mathbf{x})\|p(\mathbf{z})) = \frac{1}{2}(f_1(\sigma_1^2) + \mu_1^2 + \sum_{i=2}^{n}[f_1(\frac{\sigma_i^2}{1-\alpha^2}) + \frac{(\mu_i - \alpha\mu_{i-1})^2 + \alpha^2\sigma_{i-1}^2}{1-\alpha^2}]) \quad (3)$$

## C  DERIVATION OF THE LOWER-BOUND

Removing non-negative quadratic terms involving $\mu_i$ in equation 3 and expanding back $f$ inside the summation yields

$$D_{\mathrm{KL}}(q(\mathbf{z}|\mathbf{x})\|p(\mathbf{z})) \geq \frac{1}{2}(f_1(\sigma_1^2) + \frac{\alpha^2\sigma_1^2}{1-\alpha^2} + \sum_{i=2}^{n-1}[\frac{\sigma_i^2(1+\alpha^2)}{1-\alpha^2} - \ln(\frac{\sigma_i^2}{1-\alpha^2}) - 1] + f_1(\frac{\sigma_n^2}{1-\alpha^2}))$$

$$= \frac{1}{2}(f_{\frac{1}{1-\alpha^2}}(\sigma_1^2) + \sum_{i=2}^{n-1} f_{1+\alpha^2}(\frac{\sigma_i^2}{1-\alpha^2}) + f_1(\frac{\sigma_n^2}{1-\alpha^2}))$$

Consider $f_a(x) = ax - \ln(x) - 1$ and its first and second order derivatives, $f_a'(x) = a - \frac{1}{x}$ and $f_a''(x) \geq 0$. Thus, $f_a$ is convex and obtains its minimum value of $\ln(a)$ at $x = a^{-1}$. Substituting $\sigma_1^2 = 1 - \alpha^2$, $\sigma_n^2 = 1 - \alpha^2$ and $\sigma_i^2 = \frac{1-\alpha^2}{1+\alpha^2}$ yields the following lower-bound for the KL:

$$D_{\mathrm{KL}}(q(\mathbf{z}|\mathbf{x})\|p(\mathbf{z})) \geq \frac{1}{2}[(n-2)\ln(1+\alpha^2) - \ln(1-\alpha^2)]$$

When using multi-dimensional $\mathbf{z}_i$ at each timestep, the committed rate is the sum of the KL for each individual dimension:

$$D_{\mathrm{KL}}(q(\mathbf{z}|\mathbf{x})\|p(\mathbf{z})) \geq \frac{1}{2}[\sum_{k=1}^{D}(n-2)\ln(1+\alpha_k^2) - \ln(1-\alpha_k^2)]$$

## D  INDEPENDENT $\delta$-VAEs

The most common choice for variational families is to assume that the components of the posterior are independent, for example using a multivariate Gaussian with a diagonal covariance: $q_\phi(\mathbf{z}|\mathbf{x}) = \mathcal{N}(\mathbf{z}; \mu_q(\mathbf{x}), \sigma_q(\mathbf{x}))$. When paired with a standard Gaussian prior, $p(\mathbf{z}) = \mathcal{N}(\mathbf{z}; 0, 1)$, we can guarantee a committed information rate $\delta$ by constraining the mean and variance of the variational family (see Appendix C)

$$\mu_q^2 \geq 2\delta + 1 + \ln(\sigma_q^2) - \sigma_q^2$$

We can, thus, numerically solve

$$\ln(\sigma_q^2) - \sigma_q^2 + 2r + 1 \geq 0$$

to obtain the feasible interval $[\sigma_q^l, \sigma_q^u]$ where the above equation has a solution for $\mu_q$, and the committed rate $\delta$. Posterior parameters can thus be parameterised as:

$$\sigma_q = \sigma_q^l + (\sigma_q^u - \sigma_q^l)\frac{1}{1 + e^{-\sigma_\phi(x)}}$$
$$\mu_q = 2\delta + 1 + \ln(\sigma_q^2) - \sigma_q^2 + \max(0, \mu_\phi(\mathbf{x}))$$

Where $\phi$ parameterises the data-dependent part of $\mu_q$ ad $\sigma_q$, which allow the rate to go above the designated lower-bound $\delta$.

We compare this model with the temporal version of $\delta$-VAE discussed in the paper and report the results in Table 3. While independent $\delta$-VAE also prevents the posterior from collapsing to prior, its performance in density modeling lags behind temporal $\delta$-VAE.

| Method | Test ELBO (KL) | Accuracy |
|---|---|---|
| Independent $\delta$-VAE ($\delta = 0.08$) | 3.08 (0.08) | 66% |
| Temporal $\delta$-VAE ($\delta = 0.08$) | 3.02 (0.09) | 65% |

Table 3: Comparison of independent Gaussian delta-VAE and temporal delta-VAE with AR(1) prior on CIFAR-10 both targeting the same rate. While both models achieve a KL around the target rate and perform similarly in the downstream linear classification task, the temporal model with AR(1) prior achieves significantly better marginal likelihood.

# E  ARCHITECTURE DETAILS

## E.1  IMAGE MODELS

In this section we provide the details of our architecture used in our experiments. The overall architecture diagram is depicted in Fig. 5. To establish the anti-causal context for the inference network we first reverse the input image and pad each spatial dimension by one before feeding it to the encoder. The output of the encoder is cropped and reversed again. As show in Fig. 5, this gives each pixel the anti-causal context (i.e., pooling information from its own value and future values). We then apply average pooling to this representation to give us row-wise latent variables, on which the decoder network is conditioned.

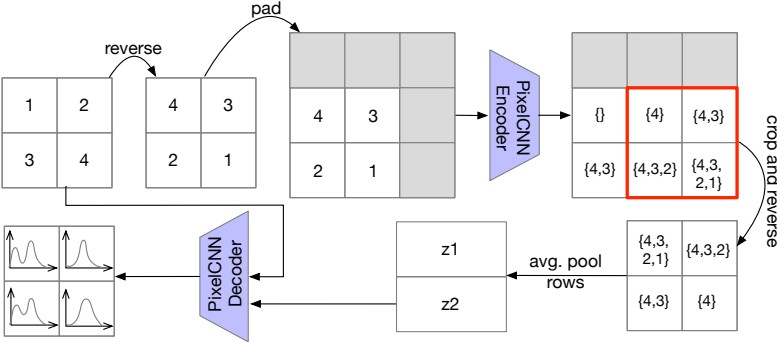

Figure 5: Architecture for images

The exact hyper-parameters of our network is detailed in Table 4. We used dropout only in our decoder and applied it the activations of the hidden units as well as the attention matrix. As in (Vaswani et al., 2017), we used rectified linear units and layer normalization (Ba et al., 2016) after the multi-head attention layers. We found layer normalization to be essential for stabilizing training. We trained with the Adam optimizer (Kingma & Ba, 2014) and used the learning rate schedule proposed in (Vaswani et al., 2017) with a few tweaks as in the following formulae:

$$LR_{imageNet} = 0.18 \times h_d^{-0.5} \min(step\_num^{0.35}, step\_num \times 4000^{1.5})$$
$$LR_{cifar10} = 0.36 \times h_d^{-0.5} \min(step\_num^{0.35}, step\_num \times 8000^{1.5})$$
$$LR_{ablation} = 0.0001$$

We use multi-dimensional latent variables per each timestep, with different slowness factors linearly spaced between a chosen interval. For our ablation studies, we chose corresponding hyper-parameters of each method we compare against to target rates between 25-100 bits per image.

| | $l_e/l_d$ | $h_e/h_d/h_{aux}$ | $r_e/r_d$ | $a_e/a_d$ | $ah$ | $n_{dmol}$ | $do_d$ | $z$ | $\alpha$ |
|---|---|---|---|---|---|---|---|---|---|
| **Best** | | | | | | | | | |
| Imagenet | 6/20 | 128/512/1024 | 1024/2048 | 2/5 | 8 | 32 | 0.3 | 16 | [0.5, 0.95] |
| CIFAR-10 | 20/30 | 128/256/1024 | 1024/1024 | 11/16 | 8 | 32 | 0.5 | 8 | [0.3, 0.99] |
| **Ablations** | | | | | | | | | |
| CIFAR-10 | 8/8 | 128/128/1024 | 1024/1024 | 2/2 | 2 | 32 | 0.2 | 32 | [0.5, 0.68-0.99] |

Table 4: Hyperparameter values for the models used for experiments. The subscripts $e$, $d$, $aux$ respectively denote the encoder, the decoder, and the LSTM auxiliary prior. $l$ is the number of layers, $h$ is the hidden size of each layer, $r$ is the size of the residual filter, $a$ is the number of attention layers interspersed with gated convolution layers of PixelCNN, $n_{dmol}$ is the number of components in the discrete mixture of logistics distribution, $do_d$ is the probability of dropout applied to the decoder, $z$ is the dimensionality of the latent variable used for each row, and the $alpha$ column gives the range of the AR(1) prior hyper-parameter for each latent.

We developed our code using Tensorflow (Abadi et al., 2016). Our experiments on natural images were conducted on Google Cloud TPU accelerators. For ImageNet, we used 128 TPU cores with batch size of 1024. We used 8 TPU cores for CIFAR-10 with batch size of 64.

### E.2 TEXT MODELS

The architecture of our model for text experiment is closely based on the Transformer network of Vaswani et al. (2017). We realize the encoder anti-causal structure by inverting the causal attention masks to upper triangular bias matrices. The exact hyper-parameters are summarized in Table 5.

| | $l$ | $h$ | $r$ | $ah$ | $d$ | $z$ | $\alpha$ |
|---|---|---|---|---|---|---|---|
| LM1B | 4 | 512 | 2048 | 8 | 0.1 | 2 | [0.2, 0.4] |

Table 5: Hyperparameter values for our LM1B experiments. $l$ is the number of layers, $h$ is the hidden size of each layer, $r$ is the size of the residual filters, $do$ is the probability of dropout, $z$ is the dimensionality of the latent variable, and the $alpha$ column gives the range of the AR(1) prior hyper-parameter for each latent dimension.

## F ABLATION STUDIES

For our ablation studies on CIFAR-10, we trained our model with the configuration listed in Table 4. After training the model, we inferred the mean of the posterior distribution corresponding to each training example in the CIFAR-10 test set, and subsequently trained a multi-class logistic regression classifier on top of it. For each model, the linear classifier was optimized for 100 epochs using the Adam optimizer with the starting learning rate of 0.003. The learning rate was decayed by a factor of 0.3 every 30 epochs.

We also report the rate-distortion curves for the CIFAR-10 training set in Fig. 6. In contrast to the graph of Fig. 4a for the test set, $\delta$-VAE achieves relatively higher negative log-likelihood on the training set compared to other methods, especially for larger rates. This suggests that $\delta$-VAE is less prone to overfitting compared to $\beta$-VAE and free-bits.

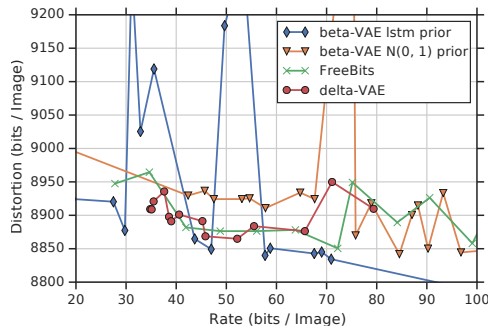

Figure 6: Rate-Distortion for CIFAR-10 training set.

## F.1 Encoder Ablation

In table Table 6, we report the details of evaluating our proposed anti-causal encoder architecture (discussed in Sect. 2.2) against the non-causal architecture in which there is no restriction on the connectivity of the encoder network. The reported experiments are conducted on the CIFAR-10 dataset. We trained 4 different configurations of our model to provide comparison in different capacity and information rate regimes, using the temporal $\delta$-VAE approach to prevent posterior collapse. We found that the anti-causal structure is beneficial when the decoder has sufficiently large receptive field, and also when encoding relatively high amount of information in latent variables.

| | $l = 6$ $h = 128$ $a = 0$ low-rate | $l = 8$ $h = 128$ $a = 2$ low-rate | $l = 8$ $h = 128$ $a = 2$ high-rate | $l_e = 20, h_e = 128$ $l_d = 30, h_d = 256$ $a = 6$ low-rate |
|---|---|---|---|---|
| Non-Causal AR(1) | 3.04 (0.02) | 3.01 (0.03) | 3.32 (0.22) | 2.88 (0.05) |
| Non-Causal Aux | 3.03 (0.01) | 2.98 (0.004) | 3.11 (0.02) | 2.85 (0.01) |
| Anti-Causal AR(1) | 3.07 (0.02) | 3.01 (0.03) | 3.22 (0.22) | 2.87 (0.05) |
| Anti-Causal Aux | 3.06 (0.01) | 2.98 (0.006) | 3.03 (0.03) | 2.84 (0.02) |

Table 6: Ablation of anti-causal vs. non-causal structure. $l$: number of layers, $h$: hidden size, $a$: number of attention layers. Subscripts $e$ and $d$ respectively denote encoder and decoder sizes when they were different. The *low-rate* (*high-rate*) models had latent dimension of 8 (64) with $alpha$ linearly placed in $[0.5, 0.95]$ ($[0.5, 0.99]$) which gives the total rate of $79.44$ ($666.6$) bits per image.

## G Visualization of the Latent Space

It is generally expected that images from the same class are mapped to the same region of the latent space. Fig. 7 illustrates the t-SNE (van der Maaten & Hinton, 2008) plot of latent variables inferred from 3000 examples from the test set of CIFAR-10 colour coded based on class labels. As can also be seen on the right hand plot, classes that are closest are also mostly the one that have close semantic and often visual relationships (e.g., cat and dog, or deer and horse).

## H Additional Samples

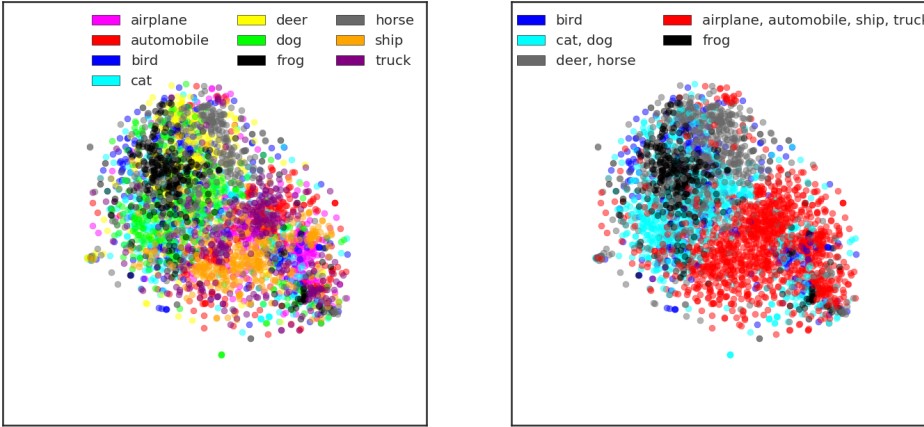

Figure 7: t-SNE plot of the posterior mean for 3000 CIFAR-10 images. Note the adjacent groups and mixed regions of the plot: cats and dogs images are mostly interspersed as are automobiles and trucks.The highest concentration of horses are on top of the region right above where deer examples are.

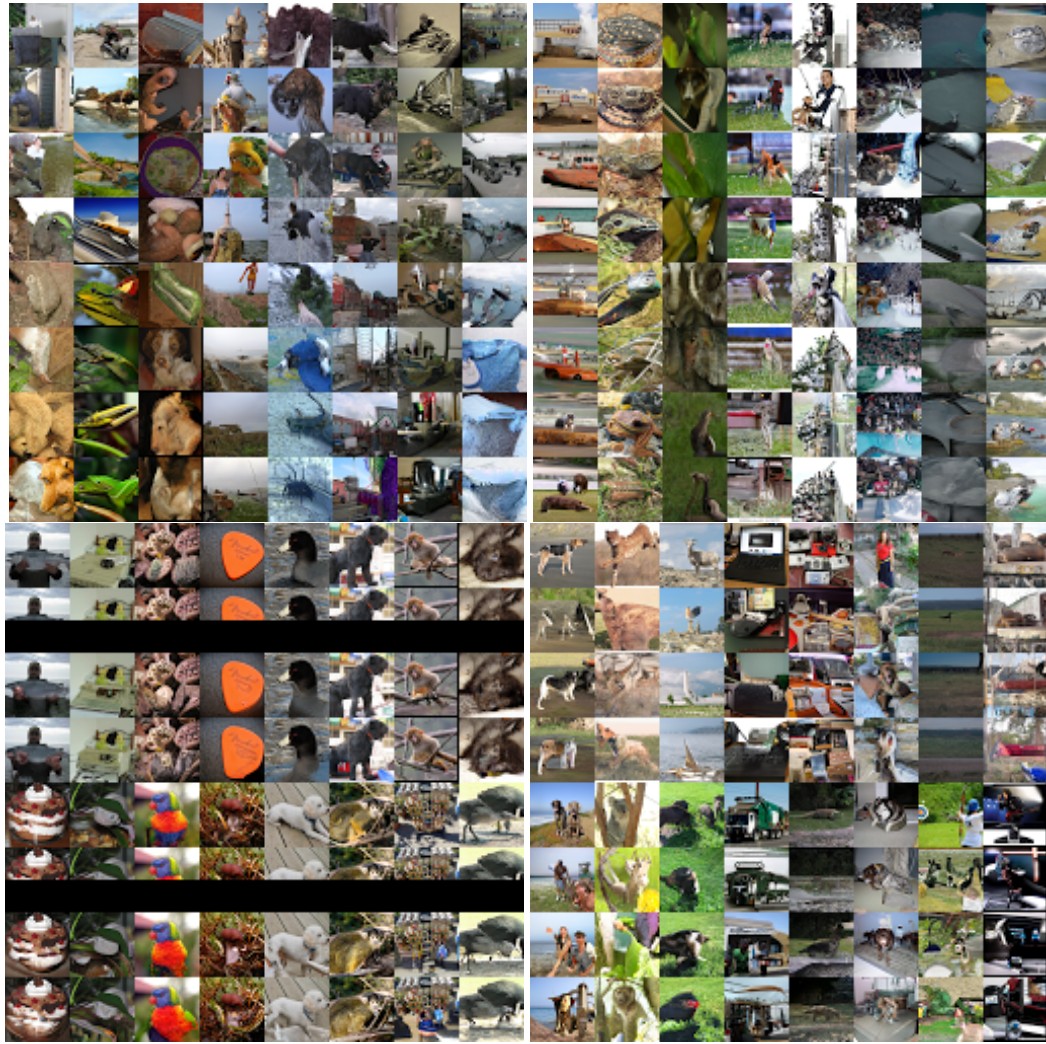

Figure 8: Additional ImageNet samples. Top left: Each column is interpolation in the latent space. Top right: "Day dream" samples where we alternate between sampling $\mathbf{x} \sim p(\mathbf{x}|\mathbf{z})$ and $\mathbf{z} \sim q(\mathbf{z}|\mathbf{x})$. Bottom left: Each half-column contains in order an original image from the validation set, occlusion of that image, and two reconstructions from different posterior samples. Bottom right: Each half-column contains in order an original image from the validation set, followed by 3 reconstructions from different posterior samples.

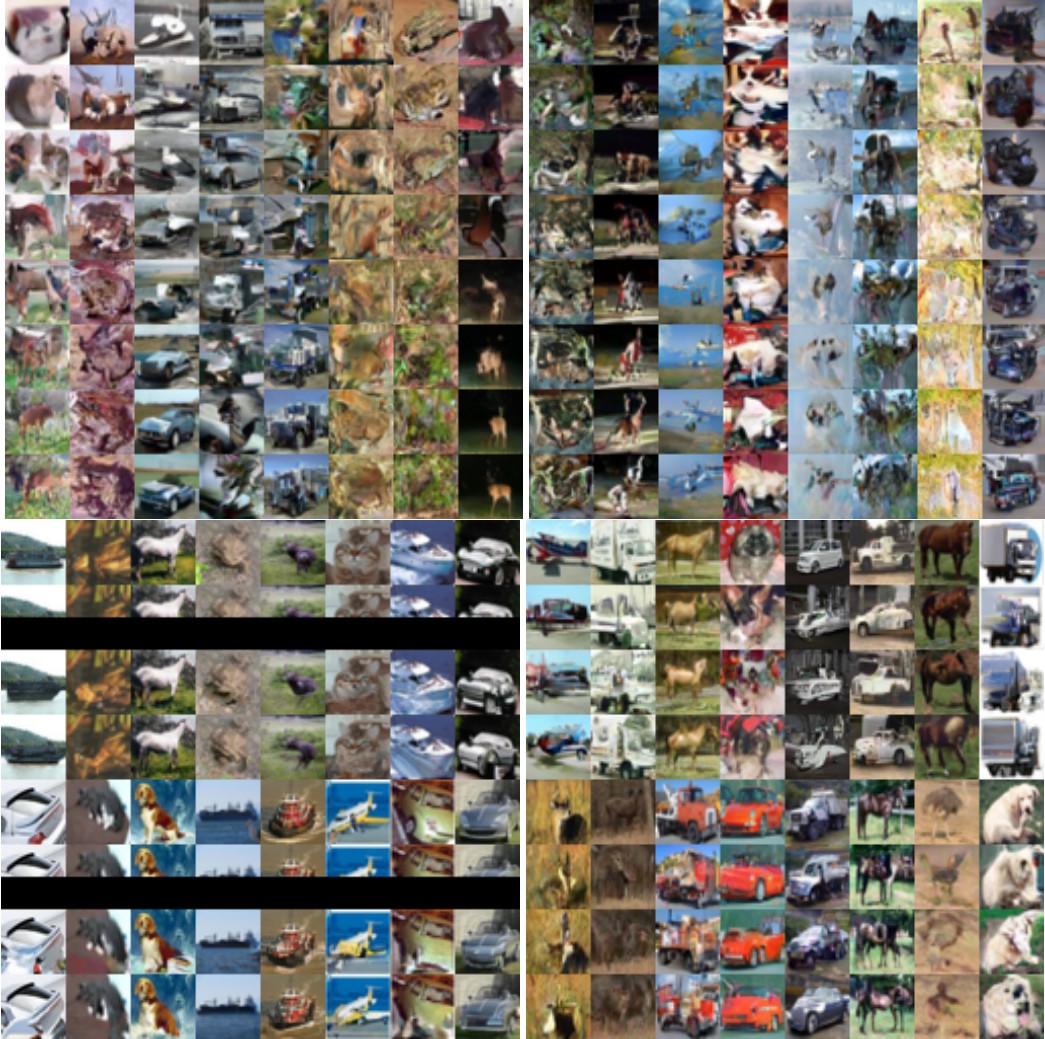

Figure 9: Additional CIFAR-10 samples. Top left: Each column is interpolation in the latent space. Top right: "Day dream" samples where we alternate between sampling $\mathbf{x} \sim p(\mathbf{x}|\mathbf{z})$ and $\mathbf{z} \sim q(\mathbf{z}|\mathbf{x})$. Bottom left: Each half-column contains in order an original image from the test set, occlusion of that image, and two reconstructions from different posterior samples. Bottom right: Each half-column contains in order an original image from the test set, followed by 3 reconstructions from different posterior samples.

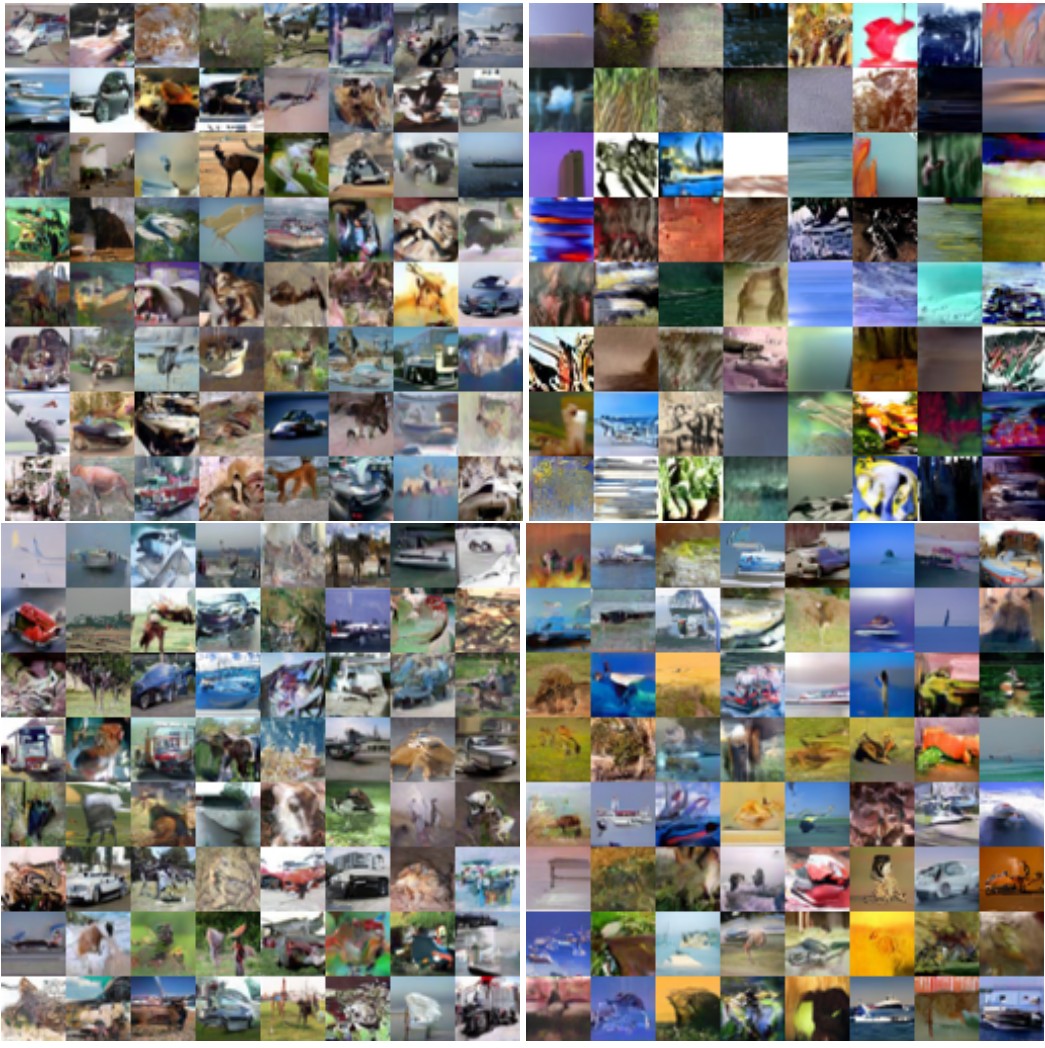

Figure 10: Random samples from the auxiliary (left) and AR(1) (right) priors of our high-rate (top) and low-rate(bottom) CIFAR-10 models. The high-rate (low-rate) model has -ELBO of 2.90 (2.83) and KL of 0.10 (0.01) bits/dim. Notice that in the high rate model that has a larger value of $\alpha$, samples from the AR(1) prior can turn out too smooth compared to natural images. This is because of the gap between the prior and the marginal posterior, which is closed by the auxiliary prior.

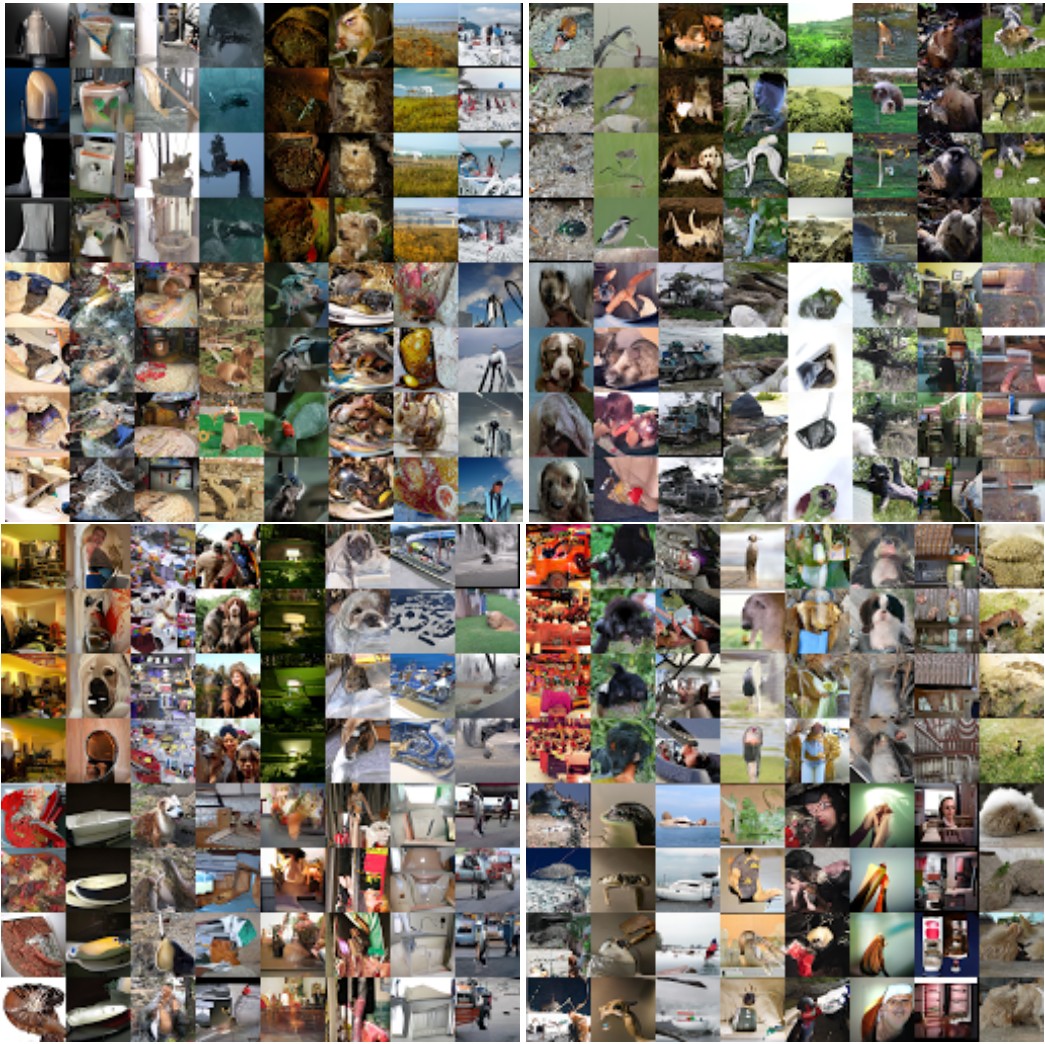

Figure 11: Additional unconditional random samples from Imagenet 32x32. Each half-column in each block contains 4 decodings of the same sample $\mathbf{z} \sim p_{aux}(\mathbf{z})$

==== Interpolating dimension 0 ====
The company's stock price is also up for a year-on-year rally, when the
The company's shares are trading at a record high for the year, when they were trading at
The company's shares were trading at $3.00, down from their 52-week low
The company's shares fell $1.14, or 5.7 percent, to $ UNK
The company, which is based in New York, said it would cut 1,000 jobs in the
The two-day meeting, held at the White House, was a rare opportunity for the United States
The company, which is based in New York, said it was looking to cut costs, but added
The company is the only company to have a significant presence in China.
The company is the only company to have a significant presence in the North American market.
The two men, who were arrested, have been released.

==== Interpolating dimension 1 ====
In the meantime, however, the company is taking the necessary steps to keep the company in the UNK
In the meantime, however, the company is expected to take some of the most aggressive steps in the
In the meantime, the company is expected to report earnings of $2.15 to $4.
The two men, who were both in their 20s , were arrested on suspicion of causing death by dangerous
The company said it was "disappointed" by a decision by the U.S. Food and Drug
The company said it would continue to provide financial support to its business and financial services clients.
The new plan would also provide a new national security dimension to U.S.- led efforts to
"I've always been a great customer and there's always there's a good chance
"It's a great personal decision...

Figure 12: One at a time interpolation of latent dimensions of a sample from the AR(1) prior. The sentences of each segment are generated by sampling a 32 element sequence of 2D random vectors from the autoregressive prior, fixing one dimension interpolating the other dimension linearly between $\mu \pm 3\sigma$.

```
==== Interpolating dimension 0 ====
"I'll be in the process of making the case," he said.
"I've got to take the best possible shot at the top," he said
"I'm not going to take any chances," he said.
"I'm not going to take any chances," he said.
"I'm not going to take any chances," he said.
We are not going to get any more information on the situation," said a spokesman for the U. N.
mission in Afghanistan, which is to be formally
We are not going to get the money back," he said.
We are not going to get the money back," said one of the co - workers.
"We are not going to get the money back," said the man.
"We are not going to get a lot of money back," said the man.

==== Interpolating dimension 1 ====
The company said the company, which employs more than 400 people, did not respond to requests for
comment, but did not respond to an email seeking comment, which
The new rules, which are expected to take effect in the coming weeks, will allow the government to
take steps to ensure that the current system does not take too
"The only thing that could be so important is the fact that the government is not going to be able to get
the money back, so the people are taking
"I'm not sure if the government will be able to do that," he said.
"We are not going to get any more information about the situation," said Mr. O'Brien.
"It's a very important thing to have a president who has a strong and strong relationship with our
country," said Mr. Obama, who has been the
"It's a very important thing to have a president who has a great chance to make a great president," said
Mr. Obama, a former senator from
"It's a very important decision," said Mr. Obama.
"It's a very difficult decision to make," said Mr. McCain.
```

Figure 13: One at a time interpolation of latent dimensions of a sample from the auxiliary prior. The generation procedure is identical to Fig. 12 with the exception that the initial vector is sampled from the auxiliary prior.

**The company is now the world's** cheapest for consumers .
**The company is now the world's** biggest producer of oil and gas, with an estimated annual revenue of $2.2 billion.

 **The company is now the world's** third-largest producer of the drug, after Pfizer and AstraZeneca, which is based in the UK.

 **The company is now the world's** biggest producer of the popular games console, with sales of more than $1bn (312m) in the US and about $3bn in the UK.
**The company is now the world's** largest company, with over $7.5 billion in annual revenue in 2008, and has been in the past for more than two decades.

 **The company is now the world's** second-largest, after the cellphone company, which is dominated by the iPhone, which has the iPhone and the ability to store in - store, rather than having to buy, the product, said, because of the Apple-based device.
**The company is now the world's** biggest manufacturer of the door-to-door design for cars and the auto industry.
**The company is now the world's** third-largest maker of commercial aircraft, behind Boeing and Airbus.
**The company is now the world's** largest producer of silicon, and one of the biggest producers of silicon in the world.
**The company is now the world's** largest maker of computer -based software, with a market value of $4.2 billion (2.6 billion) and an annual turnover of $400 million (343 million).

Figure 14: Text completion samples. For each sentence we prime the decoder with a fragment of a random sample from the validation set (shown in bold), and condition the decoder on interpolations between two samples from the latent space.

