# OpenReview forum: "Preventing Posterior Collapse with delta-VAEs"
_ICLR.cc/2019/Conference_

### Official Review · AnonReviewer2 · 2018-11-01
**Anti-causal encoder/causal decoder**

**Rating:** 6
**Confidence:** 3

**Review:**

The paper proposes a method to prevent posterior collapse, which refers to the phenomenon that VAEs with powerful autoregressive decoders tend to ignore the latent code, i.e., the decoder models the data distribution independently of the code. Specifically, the encoder, decoder, and prior distribution families are chosen such that the KL-term in the ELBO is bounded away from 0, meaning that the encoder output cannot perfectly match the prior. Assuming temporal data, the authors employ a 1-step autoregressive (across) prior with an encoder whose codes are independent conditionally on the input. Furthermore, they propose to use a causal decoder together with an anti-causal or non-causal encoder, which translates into a PixelSNAIL/PixelCNN style decoder and an anti-causal version thereof as encoder in the case of image data. The proposed approach is evaluated on CIFAR10, Imagenet 32x32, and the LM1B data set (text).

Pros:

The method obtains state-of-the-art performance in image generation. The paper features extensive ablation experiments and is well-written. Furthermore, it is demonstrated that the code learns an abstract representation by repeatedly sampling form the decoder conditionally on the code.

Cons:

One question that remains is the relative contribution of 1) lower-bounding the KL-term 2) using causal decoder/anti-causal encoder to the overall result. Is the encoder-decoder structure alone enough to prevent posterior collapse? In this context it would also be interesting to see how the encoder-decoder structure performs without \delta-constraint, but with regularization as in \beta-VAE.

What data set are the ablation experiments performed on? As far as I could see this is not specified.

Also, I suggest toning down the claims that the proposed method works "without altering the ELBO training objective" in the introduction and conclusion. After all, the encoding and decoding distributions are chosen such that the KL term in the ELBO is lower-bounded by \delta. In other words the authors impose a constraint to the ELBO.

Minor comments:
- Space missing in the first paragraph of p 5: \kappaas
- "Auxiliary prior"-paragraph on p 5: marginal posterior -> aggregate posterior?

---

> ### Author Response · Authors · 2018-11-24
> **re anti-causal structure and altered training objective**
>
> Thank you for the comments and questions!
>
> > importance of lower-bounding KL vs. encoder architecture
> We have performed additional experiments to address these questions. We found that the anti-causal encoder structure alone is not sufficient for preventing posterior collapse, while the delta-VAE alone (constraining the rate) is sufficient. Combining the anti-causal encoder with a beta-VAE objective prevents posterior collapse with small beta, but resulted in worse representations for downstream classification than delta-VAEs (see: new Figure 4).
>
> > ablations
> Ablations were performed on a smaller model on CIFAR-10. We have replaced the ablation table with a more extensive figure that shows the performance in terms of log-likelihood and linear classification accuracy for multiple techniques (beta-VAE, free bits, delta VAE in Fig. 4). We see that across all hyperparameter settings delta-VAEs results in better features for classification, and heldout ELBOs that are at least as good as other techniques.
>
> > claim of not altering the training objective
> delta-VAEs impose a *hard* constraint on the variational family, which is enforced through parameterization of the variational family. This differs from the typical soft or functional constraints that require modifying the objective and solving a constrained optimization problem using e.g. dual ascent or ALM. As we discuss in the text, by imposing hard constraints through parameterization, we do not have to alter the ELBO objectived used at training time.

---

### Official Review · AnonReviewer1 · 2018-11-01
**An interesting idea for an important problem**

**Rating:** 7
**Confidence:** 4

**Review:**

General:
The paper attacks a problem of the posterior collapse that is one of the main issues encountered in deep generative models like VAEs. The idea of the paper relies on introducing a constraint on the family of variational posteriors in such a way that the KL term could be controlled.

The authors propose to use a linear autoregressive process (AR(1)) as the prior. Alternatively, they trained a single-layer LSTM network with conditional-Gaussian outputs as the prior (the auxiliary prior). Additionally, the authors claim that the encoder should contain anti-causal dependencies in order to introduce additional bias that may diminish the posterior collapse.

The experiments present various results on image and text datasets. Interestingly, the proposed techniques allowed to perform on a par with purely autoregressive models, however, the latent variables were utilized (i.e., no posterior collapse). For instance, in Figure 3(a) we can notice that a decoder is capable of generating similar images for given latent variable. A similar situation is obtained for text data (e.g., Figure 12).

In general, I find the paper interesting and I believe it should be discussed during ICLR.

Pros:
+ The paper is well-written and all ideas are clearly presented.
+ The idea of “hard-coded” constraints is interesting and constitutes an alternative approach to utilizing either quantized values in the VAE (VQ-VAE) or a constrained family of variational posteriors (e.g., Hyperspherical VAE).
+ The obtained results are convincing. Additionally, I would like to highlight that at the first glance it might seem that there is no improvement over the autoregressive models. However, the proposed approach allows to encode an image or a document and then decode it. This is not a case for purely autoregressive models.
+ The introduction of the Slow Features into the VAE framework constitutes an interesting direction for future research.

Cons:
- The quality of Figure 4 is too low.
- I am not fully convinced that the auxiliary prior is significantly better than the AR(1) prior. Indeed, the samples seem to be a bit better for the aux. prior but it is rather hard to notice by inspecting quantitative metrics.
- In general, the proposed approach is a specific solution rather than a general framework. Nevertheless, I find it very interesting with a potential for future work.

---

> ### Author Response · Authors · 2018-11-24
> **re auxiliary prior and the generality of our approach**
>
> Thank you for your positive comments and valuable feedback!
>
> > The quality of Figure 4 is too low.
> We have improved the quality of Figure 4 and added plots of rate vs. distortion and accuracy for all techniques (beta-VAE, free bits, delta-VAE). This updated figure highlights the robustness of delta-VAEs across different hyperparameters and rates, and shows that it outperforms other approaches at all rates. This supersedes the earlier results we had in Table 1 that contained only the best achieved performance (in terms of ELBO)  for each method.
>
> > auxiliary prior
> For models that operate at higher rates, the auxiliary prior is critical to achieve SOTA performance and improve sample quality. Fig. 9 in the appendix shows that samples from the AR-1 prior are smoother and exhibit less fine-grained details than samples from the auxiliary prior. Quantitatively for our best CIFAR-10 model the difference in log-likelihood as reported per dimension does not seem large, but the auxiliary prior reduces the KL term by 72% (from 71 bits to 20 bits per image), which translates to the increased coding efficiency (i.e., reduction in distortion per transmitted bit) of  263%!
>
> > specific approach vs. framework?
> We consider the temporal and independent version of delta-VAEs as two instantiations of the general principle that the variational family should be chosen to not match the prior. Typically variational families are chosen to be maximally flexible (e.g. the work on normalizing flows), and here we present evidence that simpler and more constrained variational families are effective at regularizing generative models with rich decoders to learn more useful representations.

---

### Official Review · AnonReviewer3 · 2018-11-06
**Well written paper detailing a slightly different approach to preventing posterior collapse in VAEs.**

**Rating:** 6
**Confidence:** 3

**Review:**

The majority of approaches for preventing posterior collapse in VAEs equipped with powerful decoders to better model local structure involve either: alteration of the ELBO training objective, or a restriction on the decoder structure.

This paper presents an approach which broadly falls into the latter category; by limiting the family of the variational approximation to the posterior, the minimum KL divergence between the prior and posterior is lower bounded to a 'delta' value, preventing collapse.

The paper is well written, and the methodology clearly explained.

The experiments show that the proposed approach (delta VAE combined with the 'anti-causal' architecture) captures both local and global structure, and appears to do so while preserving SOTA discriminative performance on some tasks.  Tests are performed on both generative image and language tasks.

I believe that the paper is of low-medium significance: whilst it does outline a different method of restricting the family of posteriors, it does not give a detailed reasoning (empirical or theoretical) as to why this should be a generally better solution as compared to other approaches.

Pros:
- Very clear and well written.
- Good execution and ablation/experimentation section.

Cons:
- Lack of theory (and minimal experimentation) as to why this approach should be better than competing methods.

---

> ### Author Response · Authors · 2018-11-24
> **re evidence for the effectiveness of delta-VAEs**
>
> Thank you for your thoughtful review.
>
> > minimal experimentation
> In the original text we performed experiments on CIFAR-10, ImageNet, and LM1B to highlight the versatility of our approach. We have performed additional ablations and experiments on CIFAR-10 that shows that our proposed delta-VAE approach outperforms beta-VAE and free bits approaches for learning useful representations across a wide range of rates (Fig. 4).
>
> > lack of theory
> While we agree that different training methods may perform better in different settings, we present three reasons in the paper for why delta-VAEs may be preferable:
>
>
> 1.  Throughout the text we highlight that delta-VAEs do not require altering the training objective of the ELBO. For beta-VAEs, deviations from the ELBO at beta=1 result in an encoder, prior, and decoder that do not obey Bayes rule (Hoffman & Johnson 2016), and thus lead to worse performance in terms of log-likelihood.
>
> 2. For representation learning, the temporal-VAE approach of pairing an independent prior with a correlated prior resembles slow feature analysis which has been argued to learn more robust invariant features  (Turner & Sahani, 2007).
>
> 3. Ease of hyperparameter tuning. Given a target rate, we can analytically determine and parameterize the variational family such that the the rate is greater than or equal to the minimum target rate. This takes the form of a constraint on the mean and variances for independent delta-VAEs, and a constraint on the correlation for temporal delta-VAEs. In contrast, the relationship between beta and rate is complicated and mode- and data-dependent, thus tuning beta in beta-VAEs can be challenging. Free bits can be unstable and difficult to train, as the gradient goes from 0 to large when the constraint becomes active (see: VLAE). This motivated the authors of VLAE to use beta-VAE (which they name “soft free bits”).

---

### Public Comment · ~Adji_Bousso_Dieng1 · 2018-10-11
**Missing reference**

Hi,

Just wanted to point out our related paper https://arxiv.org/abs/1807.04863 .

---

> ### Author Response · Authors · 2018-10-12
> **RE: Missing reference**
>
> Thank you for the pointer. We will consider the paper carefully and will update our citations after the review period.

---

### Author Response · Authors · 2018-11-24
**overall response**

We thank all the reviewers for their valuable feedback. All three reviewers agree that the paper is clear and well-written. R1 and R2 highlighted the convincing results of learning useful representations with autoregressive decoders and noted our extensive experiments. R3 was concerned about experiments demonstrating the utility of our technique over other approaches (like beta-VAE and free bits), so we have added additional experiments that show delta-VAEs perform best at learning representations for downstream tasks across a large range of rates (updated Figure 4).

We believe our revised paper presents compelling evidence that delta-VAEs are a simple and effective strategy for training VAEs by constraining the parameters of the variational family to target a minimum rate. We have demonstrated improvements in log-likelihood over prior work and an ability to leverage the most recent advances in autoregressive decoders while learning latent representations that are useful for downstream tasks.

We have addressed each of their reviews in detail individually.

---

### Public Comment · ~Jaemin_Cho1 · 2018-12-21
**Missing reference for posterior collapse mitigation**

Nice work & Congrats for acceptance!
I would like to point our work which also mitigates posterior collapse :)
https://arxiv.org/abs/1804.03424

---

### Meta-Review · Area_Chair1 · 2018-12-14
**A new and not too hacky VAE training trick**

**Confidence:** 3
**Recommendation:** Accept (Poster)

**Metareview:**

Strengths:  The proposed method is relatively principled.  The paper also demonstrates a new ability: training VAEs with autoregressive decoders that have meaningful latents.  The paper is clear and easy to read.

Weaknesses:  I wasn't entirely convinced by the causal/anticausal formulation, and it's a bit unfortunate that the decoder couldn't have been copied without modification from another paper.

Points of contention:
It's not clear how general the proposed approach is, or how important the causal/anti-causal idea was, although the authors added an ablation study to check this last question.

Consensus:  All reviewers rated the paper above the bar, and the objections of the two 6's seem to have been satisfactorily addressed by the rebuttal and paper update.